# Efficient Removal of Lead, Copper and Cadmium Ions from Water by a Porous Calcium Alginate/Graphene Oxide Composite Aerogel

**DOI:** 10.3390/nano8110957

**Published:** 2018-11-20

**Authors:** Linhai Pan, Zhuqing Wang, Qi Yang, Rongyi Huang

**Affiliations:** 1AnHui Provice Key Laboratory of Optoelectronic and Magnetism Functional Materials, Anqing Normal University, Anqing 246011, China; panlinhai@nimte.ac.cn (L.P.); wangzhq@aqnu.edu.cn (Q.Y.); huangry@aqnu.edu.cn (R.H.); 2Faculty of Production Engineering, University of Bremen, Am Fallturm 1, D-28359 Bremen, Germany

**Keywords:** Adsorbent, heavy metal ions, calcium alginate, graphene oxide, polystyrene colloidal particles template

## Abstract

In this study, we fabricated a porous calcium alginate/graphene oxide composite aerogel by using polystyrene colloidal particles as sacrificial template and graphene oxide as a reinforcing filler. Owing to the excellent metal chelation ability of calcium alginate and controlled nanosized pore structure, the as-prepared calcium alginate/graphene oxide composite aerogel (mp-CA/GO) can reach the adsorption equilibrium in 40 min, and the maximum adsorption capacity for Pb^2+^, Cu^2+^ and Cd^2+^ is 368.2, 98.1 and 183.6 mg/g, respectively. This is higher than most of the reported heavy metal ion sorbents. Moreover, the mp-CA/GO can be regenerated through simple acid-washing and be used repeatedly with little loss in performance. The adsorption mechanism analysis indicates that the mp-CA/GO adsorb the heavy metal ions mainly through the ion exchange and chemical coordination effects.

## 1. Introduction

Heavy metal pollution is currently a serious environmental problem. Heavy metal ions are not biodegradable and tend to accumulate in living organisms, thereby causing many serious health and environment problems [1,2]. Therefore, it is of great significance to remove heavy metal ions in water. In recent years, many methods such as ion-exchange, chemical oxidation and reduction, membrane filtration, chemical precipitation, and adsorption have been employed for removing heavy metal ions from water media [3,4,5,6,7]. Among these treatment methods, adsorption attracts more attention due to its simple operation, low price, recyclability of the adsorbent, and high efficiency in treating low-concentration wastewater [8].

The exploration of efficient adsorbent materials starts by providing additional binding sites for adsorbed ions, improving the diffusion coefficient during the adsorption process, and enhancing the overall mechanical properties of adsorbent materials. Adsorbents with a large specific surface area, additional active groups, such as carboxyl and hydroxyl groups, and rational diffusion path for metal ion during adsorption are required. Deze et al. demonstrated the effect of porosity in heavy metal ions sorption [9].

In our previous work, we synthesized pure calcium alginate and chitosan-calcium alginate hybrid aerogels for removing heavy metal ions in water [10,11]. The results showed that calcium alginate could efficient removal of Pb^2+^ and Cu^2+^ from wastewater due to its rich carboxyl (−COOH) and hydroxyl (−OH) groups. However, dry alginate bead that has been used for heavy metal ion sorption is a rigid material without controlled porous structure, and given the low diffusion coefficient of the material, the sorption kinetics is relatively slow. Furthermore, pure alginate also exhibits several unsatisfactory structural properties, such as weak mechanical strength, structural instability, and fragile collapse [12], thus limiting their applications in the actual setting.

Graphene oxide (GO) is a new carbon material with excellent properties, high specific surface area, and rich surface functional groups. GO also exhibits a considerable potential in reinforcing fillers given its outstanding mechanical properties, excellent binding capacity, and superior flexibility [13]. The unsatisfactory properties of alginate structure and collapse of a porous material structure can be easily solved by adding GO sheets as ideal reinforcing fillers for composites. Jiao et al. presented that GO, as a reinforcing filler, exhibits excellent mechanical strength and elasticity in adsorbing heavy metal ions [13]. Recently, Yang et al. successfully prepared double network hydrogel beads by directly mixing GO and sodium alginate solutions. The as-prepared hydrogel beads showed good affinity to cationic metals and the theoretical maximum adsorption capacity for Mn^2+^ reached 56.49 mg/g [14]. However, preparing adsorbent materials with controlled pore structure and excellent mechanical properties has not been reported.

In this study, a novel macro-porous (mp) calcium alginate/graphene oxide composite aerogel (mp-CA/GO) with controlled pore structure was prepared by introducing macropores within the composite aerogel using polystyrene (PS) colloidal particles as sacrificial template and GO as reinforcing fillers. The as-prepared mp-CA/GO was characterized and used for heavy metal ions (Pb^2+^, Cu^2+^ and Cd^2+^) sorption. In addition, adsorption capacity, kinetics and thermodynamics properties, adsorption mechanism, and reutilization were also explored.

## 2. Materials and Methods 

### 2.1. Materials

Sodium alginate, ethanol, styrene, potassium persulfate (K_2_S_2_O_8_), nitric acid (HNO_3_), GO suspension (in water), sodium chloride (NaCl), and other metal salts were bought from Shanghai Aladdin Biochemical Technology Co. Ltd. (Shanghai, China). Certain reagents such as styrene, potassium persulfate, and sodium chloride, were used with further purification. High concentration GO suspension was diluted with deionized (DI) water, and DI water was used throughout this study.

### 2.2. Instruments

Scanning electron microscopy (SEM) images were obtained on a Sirion200 microscope (FEI Company, Eindhoven, the Netherlands) at an accelerating voltage of 10.0 kV. The infrared spectra were obtained from a Nicolet 6700 Fourier transform infrared (FT-IR) spectrometer (Thermo Fisher Scientific, Waltham, MA, USA). X-ray diffraction (XRD) pattern was collected on a Bruker axs D8 advanced diffractometer (Bruker Corporation, Frankfurt, Germany) using Cu Kα radiation. X-ray photoelectron spectroscopy (XPS) spectra were collected on a Shimadzu Axis-Ultra multifunctional X-ray photoelectron spectrometer (Shimadzu Corporation, Tokyo, Japan) using an Al K X-ray source. Metal concentrations were confirmed using a Perkin-Elmer Optima 2100 (Perkin-Elmer Company, Waltham, MA, USA) inductively coupled plasma-optical emission spectrometry (ICP-OES).

### 2.3. Preparation of PS Colloidal Particles

Monodisperse PS colloidal particles were synthesized by soapless emulsion polymerization [15]. Styrene is purified by vacuum distillation before using 150 mL of DI water, 0.175 g sodium chloride and 25 g styrene were added into a 250 mL three-port bottle, and stirred in the water bath at 70 °C. Then, 0.2 g of K_2_S_2_O_8_ was added to the bottle after ventilating nitrogen for nearly 20 min. With nitrogen protection, the polymerization reaction was conducted at 70 °C for 20 h. After the reaction, the particles were subjected to repeated centrifugal sedimentation at 9000 rpm and ultrasonic dispersion in water and ethanol to remove the styrene monomer and sodium chloride. Finally, the PS particles were mixed into a solution with 10% mass fraction.

### 2.4. Preparation of mp-CA/GO

For the fabrication of the mp-CA/GO, the GO (0.5 mg mL^−1^, 2 mL) and sodium alginate (1% *w*/*v*, 20 mL) solutions were mixed homogeneously. Then, PS colloidal particles solution (0.5 mL, 10 wt.%) was added to the sodium alginate and GO mixed solutions under magnetic stirring to form a homogeneous mixture. After that, this mixed solution was added dropwise to 100 mL of 0.2 M Ca^2+^ solution. A hydrogel sphere formed immediately when the mixed solution contacted with the Ca^2+^ solution. The hydrogel spheres were collected and washed with DI water, and placed into a low-temperature freezer. Then, the frozen hydrogel spheres were freeze-dried for 24 h under vacuum. The final mp-CA/GO composite aerogel were obtained by removing the PS colloidal particles with toluene and tetrahydrofuran exposure through ultrasonication [16].

### 2.5. Adsorption and Desorption Tests

In an adsorption test, ~50 mg of mp-CA/GO was added to 50 mL of 1.5 mM Pb^2+^, Cu^2+^, and Cd^2+^ solutions, respectively. The mixture was filtered after stirring for 40 min. The unextracted Pb^2+^, Cu^2+^, and Cd^2+^ in the filtrate were determined by ICP-OES.

In the desorption test, the Pb^2+^-, Cu^2+^-, or Cd^2+^-loaded mp-CA/GO was first immersed in 50 mL of 0.07 M HNO_3_ solution for 20 min. Then, the mp-CA/GO was separated and washed with DI water. The desorbed Pb^2+^, Cu^2+^, and Cd^2+^ in the eluent were also determined by ICP-OES. Each adsorption or desorption test was performed three times in parallel. The metal ion concentration was also determined three times in parallel and then averaged.

The adsorption capacity (*Q*, mg/g) and Adsorption (%) were calculated as follows:(1)$$Q=\frac{\left(C_{i}-C_{f}\right)\cdot V\cdot M}{m}$$
(2)$$\mathrm{Adsorption}=\frac{\left(C_{i}-C_{f}\right)}{C_{i}}\times 100\%$$
where *V* is the volume of the solution (L); *m* is the weight of the mp-CA/GO (g); *M* is the molar mass of metals (g mol^−1^); and *C_i_* and *C_f_* represent the initial and final concentration of metal ions in solution, respectively (mM).

## 3. Results and Discussion

### 3.1. Material Characterizations

Figure 1A illustrates the GO and PS suspensions that were used to prepare the mp-CA/GO. PS colloidal particle suspension appears milky white, whereas GO suspension appear brownish black due to high concentration. GO and PS can be dispersed in DI water and form homogeneous suspensions after a mild ultrasonic treatment. The XRD pattern of GO is presented in Figure 1D. The strong diffraction peak of GO appears at 11.5°, which can be indexed to the (002) reflections of stacked GO sheets with the interlayer distance of 0.771 nm, which allows alginate to exist between layers of GO [17,18,19]. The mp-CA/GO with excellent mechanical strength and structural stability was acquired by the help of GO. The morphology of PS and GO was characterized by SEM, as depicted in Figure 1B,C. The surface of the GO film has noticeable folds, which are attributed to the flexible, ultra-thin 2D lamination of GO (Figure 1C). As the most commonly used template material, PS colloidal particles have an excellent uniformity with a diameter of ca. 720 nm (Figure 1B), this uniformity is important for preparing materials with controlled porous structures.

The FT-IR spectra of GO, calcium alginate (CA) and mp-CA/GO are presented in Figure 2. In the three spectra, the absorption peaks around 3360 and 1412 cm^−1^ are ascribed to the O–H and –COOH, respectively. The weak peak around 2930 cm^−1^ in the spectra of CA and mp-CA/GO belongs to the asymmetric stretching vibration of C–H, in which the “egg-box” structures formed by sodium alginate macromolecule and calcium ion limit the C–H stretching [20,21]. The peaks at 3179, 1715 and 1620 cm^−1^ in the spectrum of GO are attributed to the stretching vibrations of −OH, −COOH and C=C in the sp^2^ carbon skeletal network, respectively [22]. Compared with the pure GO, the peaks of mp-CA/GO downshift from 3380 to 3346 cm^−1^ and 1612 to 1599 cm^−1^, respectively. It can be attributed to the intermolecular hydrogen bonds formed between GO sheets and alginate [23].

The SEM photos of the mp-CA/GO and CA are displayed in Figure 3. The PS beads with a diameter of ca. 720 nm are homogeneously dispersed in a composite aerogel sphere (Figure 3A), the spaces that were originally occupied by the PS beads remain as macropores after removing the particle template, and mp-CA/GO has a uniform diameter of approximately 720 nm macropore (Figure 3B). The surface appears more uneven in the mp-CA/GO with controlled pore structure than CA (Figure 3C), this phenomenon is beneficial to adsorb metal ions. In addition, the diameter of the pore structure can be regulated by changing the diameter of PS colloidal particle template, thus allowing us to design the required pore size of materials.

### 3.2. Effect of pH

In order to evaluate the effect of pH on adsorption performance, we conducted experiments at pH values that range from one to seven. As can be seen in Figure 4, adsorption capacity of the as-prepared mp-CA/GO increased significantly when the pH increased from one to three, and then remained stable with a further increase in pH. This result can be due to the change in the ionic state of the amino and carboxyl groups. When the mp-CA/GO was in strong acidic circumstance (pH < 3), functional groups (amino and carboxyl) were protonated. However, amino and carboxyl groups are deprotonated with an increase in the solution pH (3 ≤ pH ≤ 7) [10,24]. Therefore, mp-CA/GO has wide pH range to meet the needs of real application.

### 3.3. Effect of the Contact Time and Environmental Temperature

The effect of contact time on the adsorption behavior of mp-CA/GO was also evaluated (Figure 5). The adsorption rate of Pb^2+^, Cu^2+^, and Cd^2+^ onto the mp-CA/GO was quite fast, thereby completing the adsorption process within 40 min. The adsorption rate was clearly high at the initial adsorption period, possibly due to the controlled porous structures and abundant vacant sites of aerogel. The diffusion rate of metal ions was accelerated through the nanoporous structure of mp-CA/GO, thereby indicating the primary objective of preparing porous structure using PS colloidal particles as template. Furthermore, pseudo-first-order and pseudo-second-order kinetic models have been employed to investigate the adsorption kinetics during the adsorption process. The results summarized in Table 1 show that the pseudo-second-order kinetic model fits well with the kinetic data according to the values of R^2^, thus suggesting that chemical sorption is a rate determining step in the adsorption process.

Furthermore, the effect of environmental temperature on the adsorption behavior for heavy metal ions was also investigated. As seen in Figure 6, the adsorption capacity of the mp-CA/GO increased slowly with the increase in temperature. However, its improvement in terms of adsorption performance was vague, possibly because the large specific surface area and controlled porous structure of mp-CA/GO allowed the metal ions to acquire additional binding sites and rapid ion diffusion rate on the material. The adsorbent’s weak sensitivity to temperature is crucial to practical applications, thereby enabling the mp-CA/GO to be potentially applied to the practical treatment of heavy metal ions.

### 3.4. Maximum Adsorption Capacity of the mp-CA/GO

The maximum adsorption capacity is one of the most important features for the adsorbent. As seen in Figure 7, the adsorption capacity increases dramatically with the initial concentration from 0.1 mM to four mM for the mp-CA/GO. After exceeding four mM, the increment in mp-CA/GO has leveled off, possibly because of the lack of adequate functional groups to accommodate additional metal ions. The maximum adsorption capacities of mp-CA/GO for Pb^2+^, Cu^2+^, and Cd^2+^ are 368.2, 98.1 and 183.6 mg/g, respectively. These values are higher than most of the reported heavy metal ion adsorbents [25,26,27,28,29,30,31,32,33,34,35,36,37,38,39,40,41,42,43] (Table 2).

To study the adsorption behavior and predict which adsorption system is favorable. Langmuir adsorption isotherm and Freundlich adsorption isotherm models were applied in this study (Figure 8). The isotherm constants were calculated from the experimental data and are presented in Table 3. The results show that the Langmuir isotherm appears to be a favorable model to supervise the adsorption process, which indicates the adsorption process was a monolayer adsorption.

### 3.5. Adsorption Mechanism

In order to clarify the adsorption mechanism of the mp-CA/GO, we monitored the amount of each metal ion in the adsorption process. It can be seen from Figure 9 that the mole amount of Pb^2+^, Cu^2+^ and Cd^2+^ in the solution decreased while Ca^2+^ increased as the reaction proceeded. This illustrates that, in the first adsorption process, the adsorption of heavy metal ions is mainly completed by ion exchange. However, the mole amount of the adsorbed Pb^2+^, Cu^2+^, or Cd^2+^ is higher than that of the desorbed Ca^2+^ during the different adsorption periods, indicating that the ion exchange is only one of the mechanisms. In order to fully understand the essence of adsorption, we further analyzed the O one s spectra before and after Pb^2+^, Cu^2+^, and Cd^2+^ adsorption (Figure 10). The two peaks at 530.88 and 529.767 eV in the O one s spectrum are attributed to the oxygen-containing functional groups. New peaks appeared at 530.461, 530.728, and 530.866 eV, respectively, after the adsorption of Pb^2+^, Cu^2+^, and Cd^2+^, indicating that the oxygen groups of on mp-CA/GO were involved in chemisorption of Pb^2+^, Cu^2+^, and Cd^2+^. Therefore, the excellent adsorption performance of mp-CA/GO is attributed to the combined action of chemical coordination and ion exchange.

### 3.6. Regeneration Research

In the regeneration process, the mp-CA/GOs that were individually loaded with Pb^2+^, Cu^2+^, and Cd^2+^ were immersed into 50 mL of 0.07 M HNO_3_ solution. Then, the mp-CA/GO was separated and washed sequentially with DI water, calcium hydroxide solution, and then DI water till the final eluent to neutral.

The recyclability of mp-CA/GO is depicted in Figure 11. The mp-CA/GO could remove Pb^2+^, Cu^2+^ and Cd^2+^ after 20 adsorption-desorption cycles with a performance loss within five %, possibly attributed to the GO, as reinforcing fillers, to maintain the mechanical strength and elasticity of mp-CA/GO during the continuous adsorption-desorption process.

## 4. Conclusions

In this study, the mp-CA/GO with controlled pore structure was used as an efficient solid adsorbent to remove heavy metal ion in wastewater. The fabrication of a controlled porous structure has two advantages. First, the porous structure increases the surface area of the material itself, thereby enabling the adsorbent material to provide additional binding sites for heavy metal ions. The removal rates for Pb^2+^, Cu^2+^, and Cd^2+^ were 95.4%, 81.2%, and 73.2%, respectively. Second, existing porous structure significantly accelerates the diffusion rate of ions in the adsorption process, thus enabling the adsorption process to complete within 40 min. Furthermore, the mp-CA/GO could be regenerated through a simple acid washing process and used repeatedly with little loss in performance. Additionally, the wide pH application range and the weak sensitivity to temperature also allowed the mp-CA/GO to be potentially applied to actual heavy metal sewage treatment.

## Figures and Tables

**Figure 1 nanomaterials-08-00957-f001:**
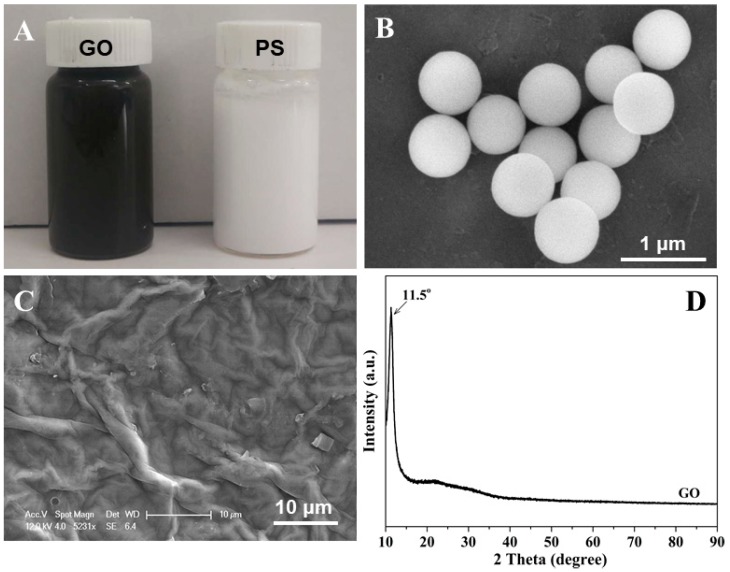
(**A**) Digital photo of GO and PS solutions, SEM images of (**B**) PS colloidal particles and (**C**) GO film, and (**D**) XRD pattern of GO.

**Figure 2 nanomaterials-08-00957-f002:**
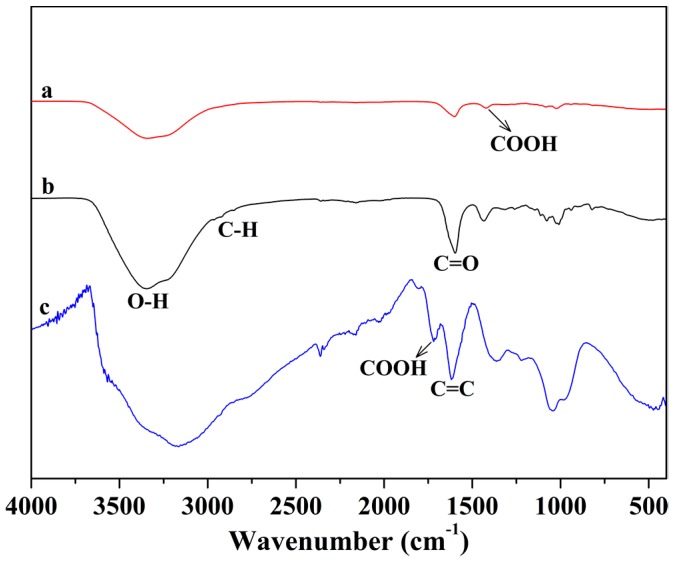
FT-IR spectra of (a) mp-CA/GO, (b) CA and (c) GO.

**Figure 3 nanomaterials-08-00957-f003:**
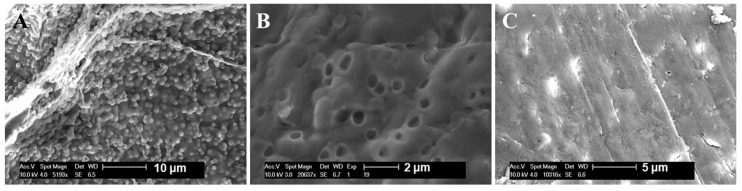
SEM photos of (**A**,**B**) mp-CA/GO/CA and (**C**) CA.

**Figure 4 nanomaterials-08-00957-f004:**
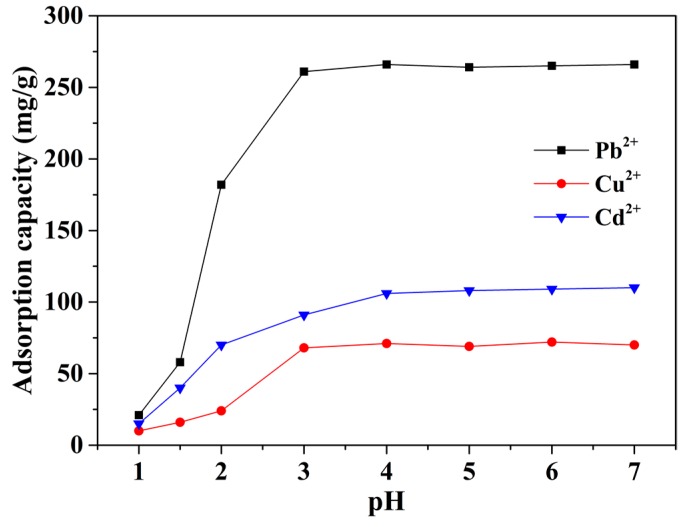
Effect of pH on metal ions adsorption. (~50 mg of mp-CA/GO was equilibrated with 50 mL of 1.5 mM Pb^2+^, Cu^2+^, or Cd^2+^ at 25 °C for 40 min).

**Figure 5 nanomaterials-08-00957-f005:**
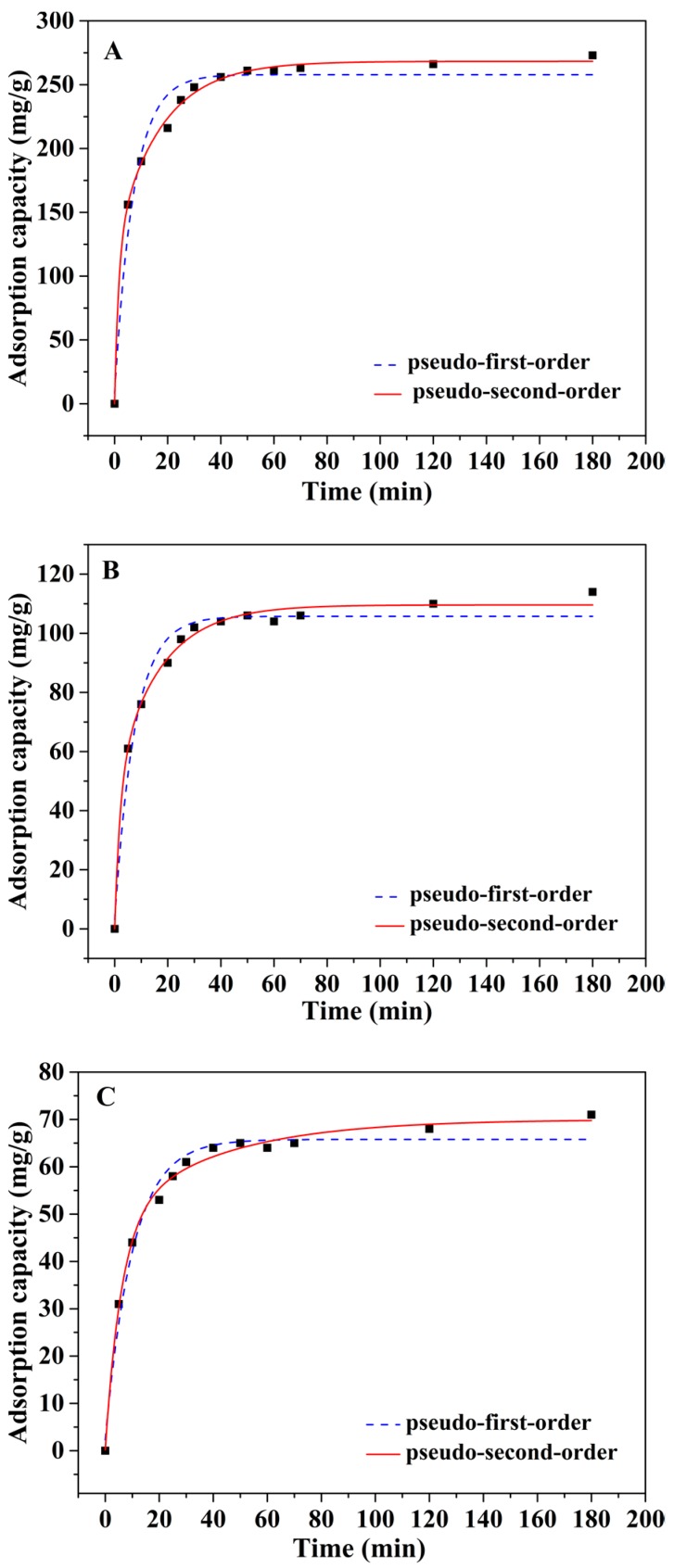
Adsorption kinetics studies of the (**A**) Pb^2+^, (**B**) Cd^2+^, and (**C**) Cu^2+^. (~50 mg of mp-CA/GO was equilibrated with 50 mL of 1.5 mM Pb^2+^, Cu^2+^, or Cd^2+^ solution at 25 °C).

**Figure 6 nanomaterials-08-00957-f006:**
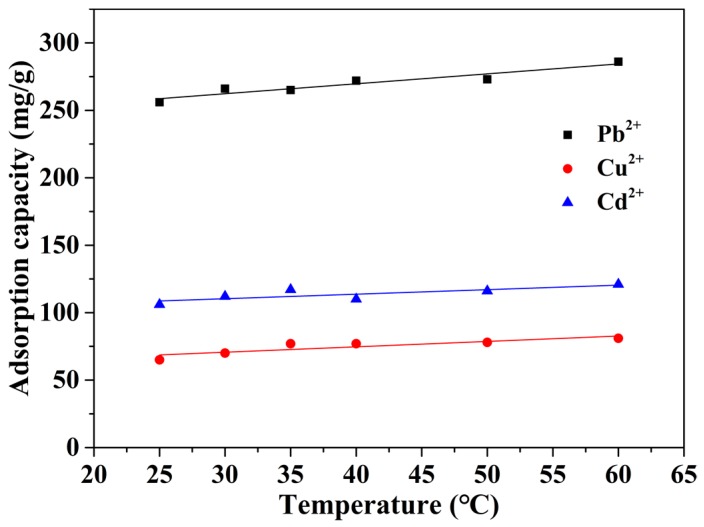
Effect of the environmental temperature on metal ions adsorption. (~50 mg of mp-CA/GO was equilibrated with 50 mL of 1.5 mM Pb^2+^, Cu^2+^, or Cd^2+^ solution at different temperatures for 40 min).

**Figure 7 nanomaterials-08-00957-f007:**
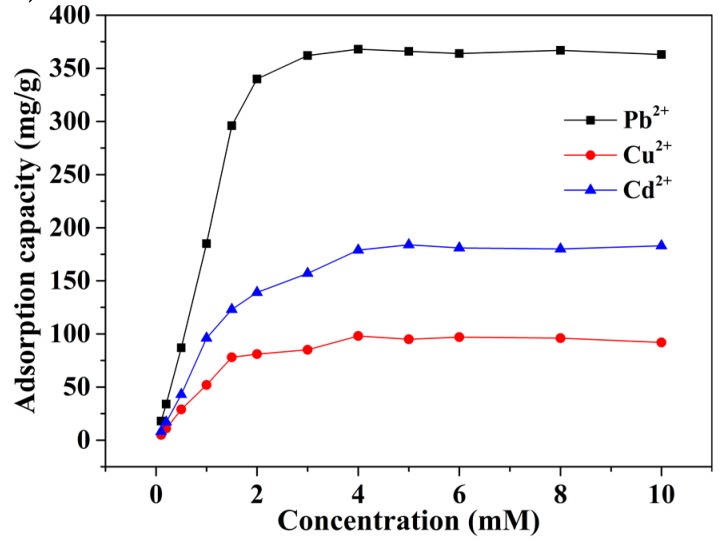
Effect of the initial metal ion concentration on adsorption. (~50 mg of mp-CA/GO was equilibrated with 50 mL of Pb^2+^, Cu^2+^, or Cd^2+^ solution at 25 °C for 40 min).

**Figure 8 nanomaterials-08-00957-f008:**
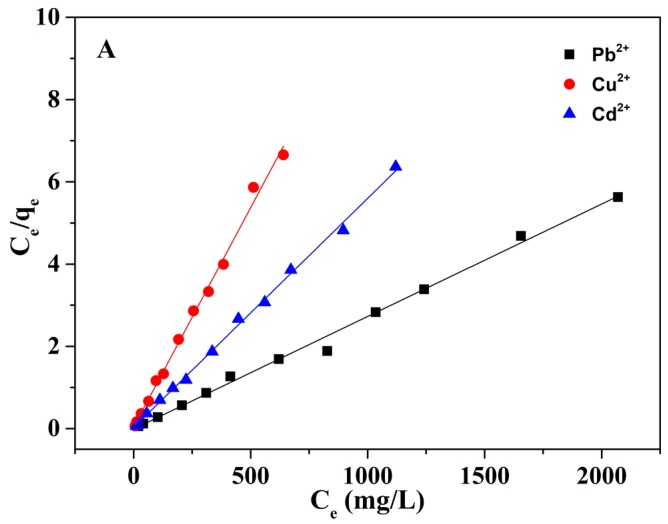
(**A**) Langmuir and (**B**) Freundlich adsorption isotherms studies of the Pb^2+^, Cd^2+^, and Cu^2+^. (~50 mg of mp-CA/GO was equilibrated with 50 mL of Pb^2+^, Cu^2+^, or Cd^2+^ solution at 25 °C for 40 min).

**Figure 9 nanomaterials-08-00957-f009:**
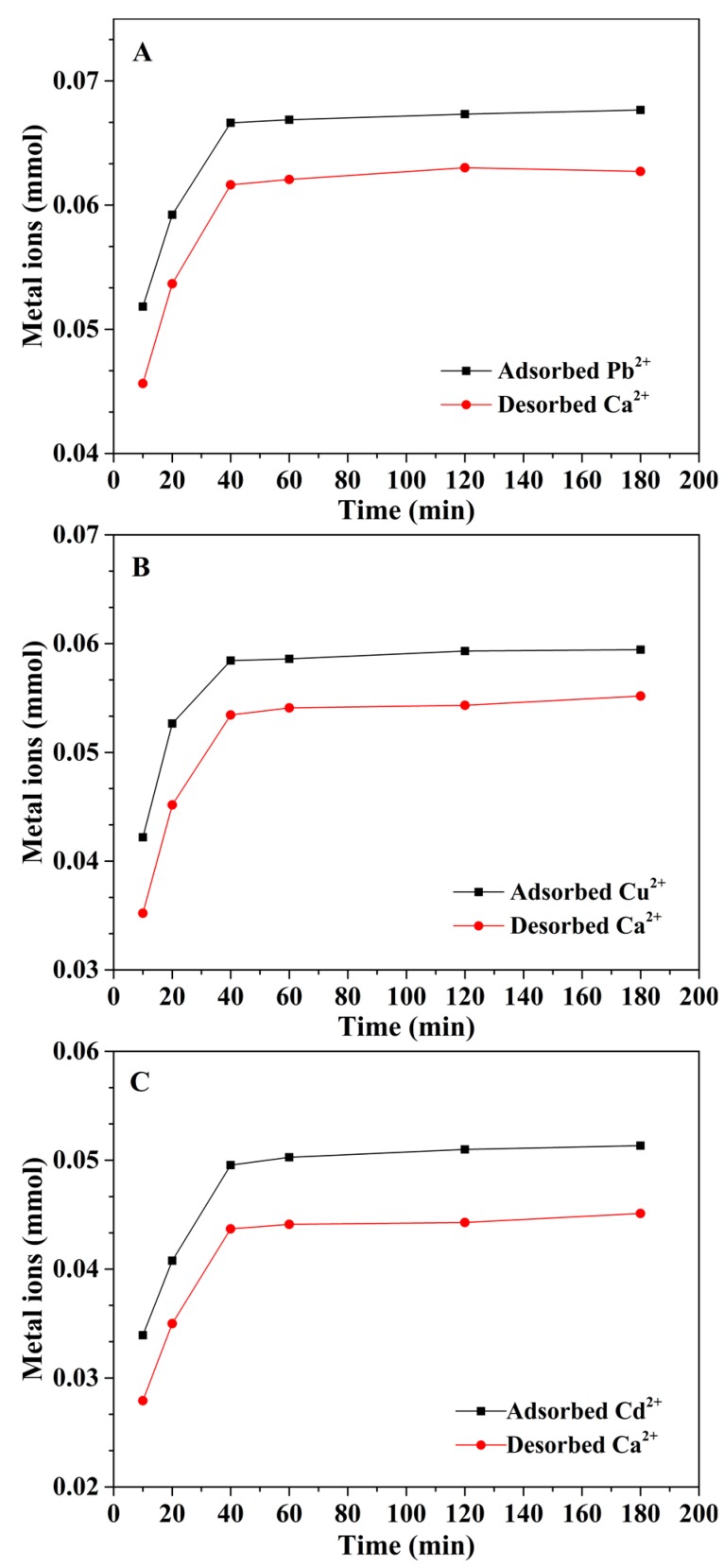
The mole amount of (**A**) Pb^2+^, Ca^2+^, (**B**) Cu^2+^, Ca^2+^ and (**C**) Cd^2+^, Ca^2+^ in different adsorption times.

**Figure 10 nanomaterials-08-00957-f010:**
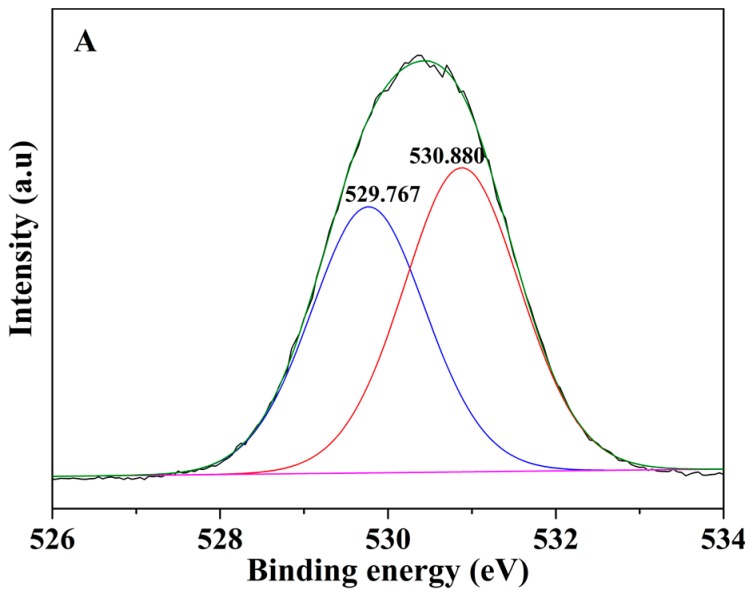
O1s XPS spectra of (**A**) mp-CA/GO, (**B**) mp-CA/GO with Pb^2+^ adsorption, (**C**) mp-CA/GO with Cu^2+^ adsorption, and (**D**) mp-CA/GO with Cd^2+^ adsorption.

**Figure 11 nanomaterials-08-00957-f011:**
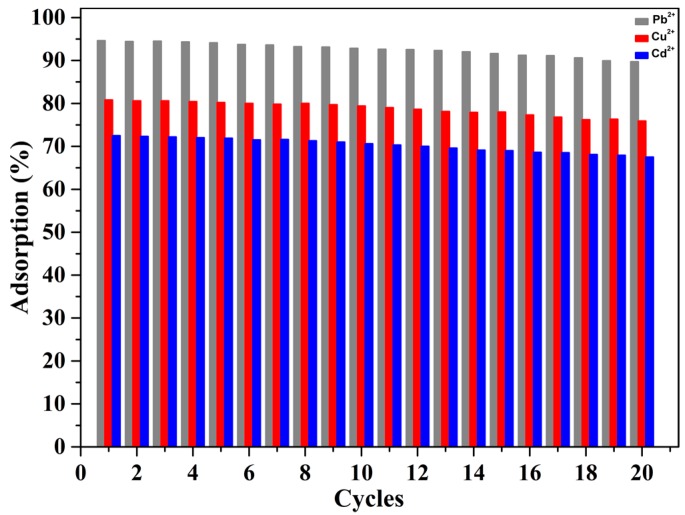
Regeneration research of mp-CA/GO. (~50 mg of mp-CA/GO and 1.5 mM of Pb^2+^ solution were used here).

**Table 1 nanomaterials-08-00957-t001:** Kinetic parameters for Pb^2+^, Cd^2+^, and Cu^2+^ ions adsorption on the mp-CA/GO.

Kinetic Model	Formula	Parameters	Pb^2+^	Cu^2+^	Cd^2+^
pseudo-first-order	q_t_ = q_e_(1 − exp(−k_1_t))	q_e_ (mg g^−1^)	257.849	65.779	105.737
		k_1_ (L min^−1^)	0.158	0.124	0.113
		R^2^	0.962	0.975	0.968
pseudo-second-order	q_t_ = q_e_(1 − 1/(1 + q_e_k_2_t))	q_e_ (mg g^−1^)	268.284	69.985	109.589
		k_2_ (L min^−1^)	0.004	0.002	0.002
		R^2^	0.996	0.992	0.991

**Table 2 nanomaterials-08-00957-t002:** Comparison of adsorption capacity of various adsorbent for Pb^2+^, Cu^2+^, and Cd^2+^.

Adsorbent	Heavy Metals (Adsorbate)	Maximum Adsorption Capacity (mg/g)	Year of Publication	Reference
Amino functionalized mesoporous silica	Pb^2+^, Ni^2+^, Cd^2+^	57.7 (Pb^2+^), 12.4 (Ni^2+^), 18.3 (Cd^2+^)	2009	[25]
Nano-alumina	Pb^2+^, Cr^3+^, Cd^2+^	100.0 (Pb^2+^), 100.0 (Cr^3+^), 83.3 (Cd^2+^)	2010	[26]
Amino functionalized magnetic graphenes composite	Pb^2+^, Hg^2+^, Cr^6+^, Cd^2+^	28.0 (Pb^2+^), 23.0 (Hg^2+^), 17.3 (Cr^6+^), 27.8 (Cd^2+^)	2014	[27]
Polydopamine microspheres	Pb^2+^	165.8	2017	[28]
Polyving alcohol/polyacrylic acid double network gel	Pb^2+^, Cd^2+^	195.0 (Pb^2+^), 115.9 (Cd^2+^)	2015	[29]
Biochar-alginate capsule	Pb^2+^	263.2	2013	[30]
Polyaniline/calcium alginate composite	Pb^2+^, Cu^2+^	357.0 (Pb^2+^), 79.0 (Cu^2+^)	2012	[31]
Silica modified calcium alginate-xanthan gum hybrid bead composite	Pb^2+^	18.9	2013	[32]
Activated carbon-calcium alginate composite	Pb^2+^	15.7	2016	[33]
Alginate-SBA-15 composite	Pb^2+^	222.2	2013	[34]
Soy protein hollow microspheres	Pb^2+^, Zn^2+^, Cr^3+^, Cd^2+^, Cu^2+^, Ni^2+^	235.6 (Pb^2+^), 255.0 (Zn^2+^), 52.9 (Cr^3+^), 120.8 (Cd^2+^), 115.0 (Cu^2+^), 177.1 (Ni^2+^)	2013	[35]
Magnetic alginate beads	Pb^2+^	50	2012	[36]
γ-Fe_2_O_3_ nanoparticles	Pb^2+^, Cu^2+^	69.0 (Pb^2+^), 34.0 (Cu^2+^)	2017	[37]
Magnetic chitosan/graphene oxide imprinted Pb^2+^	Pb^2+^	79.0	2016	[37]
Chitosan coated calcium alginate	Pb^2+^	106.9	2016	[39]
Hydroxyapatite/chitosan porous material	Pb^2+^	264.4	2015	[40]
Calcite-poly(ethyleneimine) nanostructured rod	Pb^2+^	240	2013	[41]
Nanostructured vaterite-poly(ethyleneimine) hybrid	Pb^2+^	2762	2014	[42]
Alginate-melamine hybrid	Pb^2+^	287.7	2018	[43]
mp-CA/GO	Pb^2+^, Cu^2+^, Cd^2+^	368.2 (Pb^2+^), 98.1 (Cu^2+^), 183.6 (Cd^2+^)	This work	This work

**Table 3 nanomaterials-08-00957-t003:** Isotherm parameters for Pb^2+^, Cd^2+^, and Cu^2+^ ions adsorption on the mp-CA/GO.

Isotherm Model	Formula	Parameters	Pb^2+^	Cu^2+^	Cd^2+^
Langmuir	C/q = C/q_e_ + 1/(q_e_b)	q_e_ (mg/g)	366.835	180.274	96.693
		b (L/mg)	0.493	0.473	0.415
		R^2^	0.994	0.998	0.995
Freundlich	lgq = lgK + 1/nlgC	K (L/mg)	5.263	3.863	3.573
		n	2.163	1.663	1.862
		R^2^	0.878	0.873	0.864

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
