# Peer review of "Efficient Removal of Lead, Copper and Cadmium Ions from Water by a Porous Calcium Alginate/Graphene Oxide Composite Aerogel"

_nanomaterials, 2018, doi:10.3390/nano8110957_

Round 1

Reviewer 1 Report

The ms reports on biosorbent for heavy metal removal from aqueous media. The results are interesting and they make the MS suitable for publication.  Revision is needed as detailed below.

-          The adsorption isotherms (fig 7) should be quantitatively analysed. This would provide the affinity (equilibrium constant) for the adsorption process beside the maximum adsorption capacity that is discussed in the MS.

-          Table 1. The number of digits for the obtained parameters should be revised based on the error from the fitting. There are many decimal digits that are certainly not significant.

-          Introduction is not up-to-date and recent studies on biosorbents should be added (see: Appl. Clay Sci. 2013, 72, 132–137, doi:10.1016/j.clay.2012.12.001; Appl. Sci. 2018, 8(9), 1518; https://doi.org/10.3390/app8091518; Nanomaterials 2017, 7(3), 57; https://doi.org/10.3390/nano7030057; Appl. Clay Sci. 156(156), 87-95)

-          Minor point. Equations should be numbered.

Author Response

The ms reports on biosorbent for heavy metal removal from aqueous media. The results are interesting and they make the MS suitable for publication. Revision is needed as detailed below.

Answer: Thanks for your nice comments here.

Question 1: The adsorption isotherms (fig 7) should be quantitatively analysed. This would provide the affinity (equilibrium constant) for the adsorption process beside the maximum adsorption capacity that is discussed in the MS.

Answer: According to your suggestion, the adsorption isotherm models (Langmuir and Freundlich models) have been employed to fitting the experimental data (Figure 8), and the analysis results have also been added to section of “3.4. Maximum adsorption capacity of the mp-CA/GO”.

Question 2: Table 1. The number of digits for the obtained parameters should be revised based on the error from the fitting. There are many decimal digits that are certainly not significant.

Answer: Thanks for your kindly reminding. The sensitivity of the experimental data is 3 decimal numbers. In order to make the parameters values consistent with the experimental data, the calculated kinetic parameters should also be retained 3 decimal numbers. Thus, we corrected the decimal number of kinetic parameters in Table 1.

Question 3: Introduction is not up-to-date and recent studies on biosorbents should be added (see: Appl. Clay Sci. 2013, 72, 132–137, doi:10.1016/j.clay.2012.12.001; Appl. Sci. 2018, 8(9), 1518; https://doi.org/10.3390/app8091518; Nanomaterials 2017, 7(3), 57; https://doi.org/10.3390/nano7030057;Appl. Clay Sci. 156(156), 87-95).

Answer: Following your kind suggestion, these related references [2], [5], [8] and [43] have been cited and added to the revised manuscript.

2.     Voisin, H.; Bergstrom, L.; Liu, P.; Mathew, A.P. Nanocellulose-based materials for water purification. Nanomaterials 2017, 7, 53.

5.     Cataldo, S.; Lazzara, G.; Massaro, M.; Muratore, N.; Pettignano, A.; Riela, S. Functionalized halloysite nanotubes for enhanced removal of lead(II) ions from aqueous solutions. Appl Clay Sci 2018, 156, 87-95.

8.     Cavallaro, G.; Gianguzza, A.; Lazzara, G.; Milioto, S.; Piazzese, D. Alginate gel beads filled with halloysite nanotubes. Appl Clay Sci 2013, 72, 132-137.

43.  Li, K.T.; Wu, G.H.; Wang, M.; Zhou, X.H.; Wang, Z.Q. Efficient removal of lead ions from water by a low-cost alginate-melamine hybrid sorbent. Applied Sciences 2018, 8, 1518.

Question 4: Minor point. Equations should be numbered.

Answer: The equations have been numbered in the revised manuscript. Please see page 3, line 20. Thanks again.

Reviewer 2 Report

Nanomaterials 390972

Title: “Efficient removal of lead, copper and cadmium ions from wáter by a porous calcium alginate/Graphene oxide composite aerogel”

In this paper, the authors present the synthesis a porous calcium alginate/graphene oxide composited aerogel, using polystyrene colloidal, graphene oxide and calcium alginate (mp-CA/GO), obtaining maxima absorption for Pb2+, Cu2+ and Cd2+. Moreover, the (mp-CA/GO) can be regenerated through acid washing. The study indicate that mp-CA/GO absorb the heavy metal ions.

In my opinion even though correctly performed. The chemistry of this paper is sufficiently interesting, but is necessary one major revision of all manuscript.

Comment:

1)      Complete the references. For example,

a)      J. Mat. Chem. A, 2014, 2, 8766-8772

b)      J. Mat. Chem. A, 2013, 1, 13532-13541

2)      Page 2, Line 21

Add     NaCl

3)      Page 2, Line 35

CHANGE      using.              FOR    using

4)      Page 2, Line 41

Define the letters: PS

5)      Page 3, Line 22

CHANGE      Go                   FOR    GO

6)      Page 3, Line 30

CHANGE      ca.720 nm       FOR    ca. 720 nm

7)      Page 4, Line    4

CHANGE      exhibited         FOR    presented

Revise all manuscript

8)      Page 4, Line 4-12

Revise the assignment and description of the IR spectra

9)      Page 5, Line 13-15

Revise and re-write the paragraph “As seen in Figure 4, ·····further pH increasement”

10)  Page 8, Line 10

CHANGE      O 1s spectrum            FOR    O1s spectra

Revise all manuscript

Author Response

Title: “Efficient removal of lead, copper and cadmium ions from wáter by a porous calcium alginate/Graphene oxide composite aerogel”. In this paper, the authors present the synthesis a porous calcium alginate/graphene oxide composited aerogel, using polystyrene colloidal, graphene oxide and calcium alginate (mp-CA/GO), obtaining maxima absorption for Pb2+, Cu2+ and Cd2+. Moreover, the (mp-CA/GO) can be regenerated through acid washing. The study indicate that mp-CA/GO absorb the heavy metal ions. In my opinion even though correctly performed. The chemistry of this paper is sufficiently interesting, but is necessary one major revision of all manuscript.

Answer: Thanks for your comments here.

Question 1): Complete the references. For example, a) J. Mat. Chem. A, 2014, 2, 8766-8772. b) J. Mat. Chem. A, 2013, 1, 13532-13541.

Answer: According to your nice suggestion, the relevant references [41] and [42] have been cited and added to the revised manuscript.

41.  Marzo, A.M.L.; Pons, J.; Merkoci, A. Multifunctional system based on hybrid nanostructured rod formation, for sensoremoval applications of Pb2+ as a model toxic metal. J Mater Chem A 2013, 1, 13532-13541.

42.  Lopez-Marzo, A.M.; Pons, J.; Merkoci, A. Extremely fast and high Pb2+ removal capacity using a nanostructured hybrid material. J Mater Chem A 2014, 2, 8766-8772.

Questions: 2)  Page 2, Line 21 Add NaCl; 3) Page 2, Line 35, CHANGE using. FOR using; 4) Page 2, Line 17, Define the letters: PS; 5) Page 3, Line 22. CHANGE Go FOR GO; 6) Page 3, Line 30, CHANGE ca.720 nm FOR ca. 720 nm; 7) Page 4, Line 4. CHANGE exhibited FOR presented.

Answer: In the revised manuscript, we have corrected these contents following your suggestions.

Question 8): Page 4, Line 4-12. Revise the assignment and description of the IR spectra.

Answer: In the revised manuscript, we have corrected these contents following your suggestion. Please see page 4, line 4 to16.

Question 9): Page 5, Line 13-15. Revise and re-write the paragraph “As seen in Figure 4, ·····further pH increasement”.

Answer: According to your kind suggestion, this sentence has been rephrased in the new manuscript. Please see page 5, line 16 to 17.

Question 10): Page 8, Line 10. CHANGE O 1s spectrum FOR O1s spectra.

Answer: “O 1s spectrum” has been revised as “O 1s spectra” in the new manuscript. Thanks.

Reviewer 3 Report

This manuscript presents a self-contained work related to metal ions removal from water based on an alginate/GO composite. The authors may consider the following comments to highlight their achievements:

Previous works have already reported the absorption of metal by Sodium Alginate/GO (i.e. Scientific Reports, volume 8, Article number: 10717 (2018). The authors should clearly state the benefits of their Calcium Alginate /GO approach with respect to that.

Table 2 and related discussion can be completed by including the recent work of Li et al., “Efficient Removal of Lead Ions from Water by a Low-Cost Aliganate-Melamine Hybrid Sorbent”.

It will be interesting also if the authors include the year of publication within this table so the potential readers can have a clear overview of the evolution of the State of the Art. Please also add “this work” with your achievements: (Pb 2+ , Cu 2+ and Cd 2+, 368.2, 98.1 and 183.6 mg/g)

Regeneration and reusability are key factors in improving wastewater process economics. Figure 10 provides little information in this regard. You may magnify the scale in the percentage of interest to highlight the actual trend or increase the number of cycles to estimate the actual projection.

There are some grammatical mistakes and errors along the text. I recommend careful revision. To name a few of them:

Pag.3. Lin. 21 appear → appears

Pag.3. Lin. 33 Go → GO

Pag.3. Lin. 30 particle → particles

Pag.6. Lin. 2 shows → show

Pag.5. Lin. 6 were → was

Author Response

Question 1: This manuscript presents a self-contained work related to metal ions removal from water based on an alginate/GO composite. The authors may consider the following comments to highlight their achievements: Previous works have already reported the absorption of metal by Sodium Alginate/GO (i.e. Scientific Reports, volume 8, Article number: 10717 (2018). The authors should clearly state the benefits of their Calcium Alginate /GO approach with respect to that.

Answer: Thanks for your comments here. There are two differences between our work and the work of Yang et al. The first is that during the preparation of the adsorbent, we introduced polystyrene (PS) colloidal particles, which can be used as a sacrificial template to form a controlled pore size structure. The second difference is that Yang et al.'s work focuses on the adsorption of manganese ions (Mn2+), and our work is to adsorb toxic heavy metals (such as Pb2+ and Cd2+).

Following your nice suggestion, Yang and coworkers’ paper has been cited and introduced in section of “Introduction”, and also added to the references.

14.  Yang, X.Z.; Zhou, T.Z.; Ren, B.Z.; Hursthouse, A.; Zhang, Y.Z. Removal of Mn(II) by sodium alginate/graphene oxide composite double-network hydrogel beads from aqueous solutions. Sci Rep 2018, 8, 10717.

Question 2: Table 2 and related discussion can be completed by including the recent work of Li et al., “Efficient Removal of Lead Ions from Water by a Low-Cost Aliganate-Melamine Hybrid Sorbent”.

Answer: According to your nice suggestion, this new published literature was cited and the relevant information was also presented in Table 2.

43.  Li, K.T.; Wu, G.H.; Wang, M.; Zhou, X.H.; Wang, Z.Q. Efficient removal of lead ions from water by a low-cost alginate-melamine hybrid sorbent. Applied Sciences 2018, 8, 1518.

Question 3: It will be interesting also if the authors include the year of publication within this table so the potential readers can have a clear overview of the evolution of the State of the Art. Please also add “this work” with your achievements: (Pb2+, Cu2+ and Cd2+, 368.2, 98.1 and 183.6 mg/g).

Answer: Following your kind suggestion, the year information of the literature has been added into Table 2.

Question 4: Regeneration and reusability are key factors in improving wastewater process economics. Figure 10 provides little information in this regard. You may magnify the scale in the percentage of interest to highlight the actual trend or increase the number of cycles to estimate the actual projection.

Answer: Following your kind suggestion, we have increased the number of cycles of mp-CA/GO and the experimental results have also been added to Figure 11.

Question 5: There are some grammatical mistakes and errors along the text. I recommend careful revision. To name a few of them: Pag.3. Lin. 21 appear → appears; Pag.3. Lin. 33 Go → GO; Pag.3. Lin. 30 particle → particles; Pag.6. Lin. 2 shows → show; Pag.5. Lin. 6 were → was.

Answer: In the revised manuscript, we have corrected these words following your suggestions. Thanks again.

Reviewer 4 Report

The paper reports on the development of a new method for the efficient removal of heavy metal ions from wastewater. The method is based on the porous calcium alginate/graphene oxide composite aerogel (mp-CA/GO). The Authors have utilized polystyrene (PS) colloidal particles as the sacrificial template and GO as the reinforcing filler. The obtained materials have been characterized by SEM images, XRD pattern and FT-IR spectra. The influence of pH, contact time, and the environmental temperature on metal ions adsorption behavior of mp-CA/GO, have also been investigated. The adsorption kinetics and mechanism have been elucidated. A regeneration procedure for the adsorbent material was also developed.

I recommend the paper for publication after minor revisions addressing the comments presented below.

1.      The meaning of "mp" in mp-CA/GO has not been defined. Does it stand for "micro-porous" ?

2.      In Figures 4, 7, and 8, the determination errors should be specified, e.g. by stating the std. or providing error bars at each data point. Fitting curves to the experimental data points, instead of the points' connections, would improve the presentation.

3.      Figure captions should list more details concerning the experimental conditions, such as the concentration of ions, pH, solution concentration, temperature, etc.

4.      The solvents used in solutions of sodium alginate and colloidal PS solution are not provided.

5.      In discussion of FTIR spectra, the term "adsorption peak" (page 4, line 5) should be replaced with "absorption peak".

6.      Many methods based on novel adsorbent materials have recently been proposed. For instance, a method based on aggregation of functionalized nanoparticles by heavy metal ions has recently been published (Hepel, M.; Blake, D.; McCabe, M.; Stobiecka, M.; Coopersmith, K. Assembly of gold nanoparticles induced by metal ions. In Functional Nanoparticles for Bioanalysis, Nanomedicine & Bioelectronic Devices Vol. 1; Hepel, M., Zhong, C.J. [Eds.]; Oxford University Press: Oxford, 2012; Vol. 1112; pp 207-240; DOI: 10.1021/bk-2012-1112.ch008). Doped polypyrrole films have also been proposed for heavy metal remediation (Electroanalysis 1996, 8, 996-1005; Microchemical Journal 1997, 56, 79-92).  These pertinent literature references should be cited.

Author Response

The paper reports on the development of a new method for the efficient removal of heavy metal ions from wastewater. The method is based on the porous calcium alginate/graphene oxide composite aerogel (mp-CA/GO). The Authors have utilized polystyrene (PS) colloidal particles as the sacrificial template and GO as the reinforcing filler. The obtained materials have been characterized by SEM images, XRD pattern and FT-IR spectra. The influence of pH, contact time, and the environmental temperature on metal ions adsorption behavior of mp-CA/GO, have also been investigated. The adsorption kinetics and mechanism have been elucidated. A regeneration procedure for the adsorbent material was also developed. I recommend the paper for publication after minor revisions addressing the comments presented below.

Answer: Thanks for your nice comments here.

Question 1: The meaning of "mp" in mp-CA/GO has not been defined. Does it stand for "micro-porous" ?

Answer: The “mp” is an abbreviation for “macro-porous”. Following your nice suggestion, the meaning of “mp” has been added to the revised manuscript.

Question 2: In Figures 4, 7, and 8, the determination errors should be specified, e.g. by stating the std. or providing error bars at each data point. Fitting curves to the experimental data points, instead of the points' connections, would improve the presentation.

Answer: In this study, each adsorption or desorption test was performed three times in parallel. Therefore, the metal ion concentration was also measured three times in parallel and then averaged. Following your nice suggestion, the relevant description has been added to section of “2.5. Adsorption and desorption tests”.

Furthermore, adsorption kinetic and isotherm models have also been employed to fitting the experimental data (Figures 5, 6 and 8), and the analysis results have been added to sections of “3.3. Effect of the contact time and environmental temperature” and “3.4. Maximum adsorption capacity of the mp-CA/GO”.

Question 3: Figure captions should list more details concerning the experimental conditions, such as the concentration of ions, pH, solution concentration, temperature, etc.

Answer: The information of experimental conditions has been added in the caption of the Figures.

Question 4: The solvents used in solutions of sodium alginate and colloidal PS solution are not provided.

Answer: The solvents used in solutions of sodium alginate and colloidal PS solution are both deionized (DI) water. Following your suggestion, the corresponding information has been added in section of “2.1. Materials”.

Question 5: In discussion of FTIR spectra, the term "adsorption peak" (page 4, line 5) should be replaced with "absorption peak".

Answer: In the new manuscript, the “adsorption peak” has been revised as “absorption peak”.

Question 6: Many methods based on novel adsorbent materials have recently been proposed. For instance, a method based on aggregation of functionalized nanoparticles by heavy metal ions has recently been published (Hepel, M.; Blake, D.; McCabe, M.; Stobiecka, M.; Coopersmith, K. Assembly of gold nanoparticles induced by metal ions. In Functional Nanoparticles for Bioanalysis, Nanomedicine & Bioelectronic Devices Vol. 1; Hepel, M., Zhong, C.J. [Eds.]; Oxford University Press: Oxford, 2012; Vol. 1112; pp 207-240; DOI: 10.1021/bk-2012-1112.ch008). Doped polypyrrole films have also been proposed for heavy metal remediation (Electroanalysis 1996, 8, 996-1005; Microchemical Journal 1997, 56, 79-92). These pertinent literature references should be cited.

Answer: According to your nice suggestion, these related references [6], [7] and [21] have been cited and added to the revised manuscript.

6.     Hepel, M.; Dentrone, L. Controlled incorporation of heavy metals from aqueous solutions and their electrorelease using composite polypyrrole films. Electroanal 1996, 8, 996-1005.

7.     Hepel, M.; Zhang, X.M.; Stephenson, R.; Perkins, S. Use of electrochemical quartz crystal microbalance technique to track electrochemically assisted removal of heavy metals from aqueous solutions by cation-exchange composite polypyrrole-modified electrodes. Microchem J 1997, 56, 79-92.

21. Maria, H.; Dustin, B.; Matthew, M.; Magdalena, S.; Kaitlin, C. Assembly of Gold Nanoparticles Induced by Metal Ions. In Functional Nanoparticles for Bioanalysis, Nanomedicine, and Bioelectronic Devices; Hepel, M., Zhong, C.J., Eds.; ACS Symposium Series, Oxford University Press: Oxford, UK, 2012; Volume 1, pp. 207–240.

Round 2

Reviewer 1 Report

MS was improved upon revision

Reviewer 2 Report

This paper is accept in present  form